# Overview of Phytochemical Composition of *Brassica oleraceae* var. *capitata* Cultivars

**DOI:** 10.3390/foods13213395

**Published:** 2024-10-25

**Authors:** Olga Statilko, Thalia Tsiaka, Vassilia J. Sinanoglou, Irini F. Strati

**Affiliations:** Laboratory of Chemistry, Analysis & Design of Food Processes, Department of Food Science and Technology, University of West Attica, Agiou Spyridonos, 12243 Egaleo, Greece; ostatilko@uniwa.gr (O.S.); tsiakath@uniwa.gr (T.T.); vsina@uniwa.gr (V.J.S.)

**Keywords:** *Brassica oleraceae* var. *capitata*, white cabbage, red cabbage, phytochemical composition, cultivars, glucosinolates

## Abstract

The Brassicaceae family includes a wide range of horticultural crops of economic and traditional importance, consumed either fresh, cooked, or fermented. Cabbage (*Brassica oleraceae* var. *capitata*) is one of the most important crops of the family. The present review analyzes the most important phytochemicals present in cabbage, focusing on variation of phytochemical composition between cultivars of *B. oleraceae* var. *capitata* f. *alba*, *B. oleraceae* var. *capitata* f. *rubra*, *B. oleraceae* var. *capitata* f. *acuta*, and *B. oleraceae* var. *capitata* f. *sabauda*. Cabbage form and cultivars significantly affect phytochemical compositions. *B. oleraceae* var. *capitata* f. *rubra* cultivars are generally great sources of phenolic compounds, especially anthocyanins, whereas *B. oleraceae* var. *capitata* f. *alba* cultivars display the highest concentration of glucosinolates; nevertheless, their levels are also dependent on the specific cultivar. *B. oleraceae* var. *capitata* f. *acuta* cultivars may be considered advantageous due to their high glucosinolate content and consistent phytochemical composition. Recognizing the benefits of specific cultivars can be valuable for consumers seeking a healthier lifestyle, as well as for scientists aiming to enhance cultivars through breeding programs or use plants’ extracts to produce high quality pigments and dietary supplements.

## 1. Introduction

The Brassicaceae (syn. Cruciferae), commonly known as the ‘mustard family’, is the largest family within the Brassicales plant order, consisting of 4636 known species and 340 genera [1]. *Brassica oleraceae* is a morphologically diverse species, that includes several common vegetables, notably cabbage, kale, broccoli, and cauliflower. Cabbage (*Brassica oleraceae *var. *capitata*) belongs to the Capitata group of the species. As the name suggests, leaves are formed into characteristic cabbage heads (‘capita’ = head) which vary in shape, color, and leaf texture, resulting in a great number of different cultivars [2]. The Capitata group consists of four different forms (Figure 1), namely *alba* (white cabbage), *rubra* (red cabbage), *sabauda* (Savoy cabbage), and *acuta* (conical cabbage), varying in their morphological characteristics [3]. More specifically, white cabbages display a smooth leaf texture that forms tight round or flat heads that are green, or pale green to white in color; pointed cabbages also display smooth textures but form cone-shaped heads; Savoy cabbages present rough leaf textures and form looser heads; red cabbages form compact heads with often oblong shapes and are characterized by their purple color [4,5].

Plant-based foods are a rich source of bioactive compounds, including phytochemicals, which exert beneficial effects on human health. Phytochemicals are secondary metabolites of low molecular weight that occur naturally in plants as a response to biotic and abiotic stresses, along with pollinator attraction [6]. They possess antioxidant activity, and exhibit antimicrobial, antidiarrheal, antiallergenic, antiviral, and anticonvulsant properties [7,8,9,10,11,12]. Phytochemicals have also been implicated in reduced risk of cancer [13,14,15]. Brassica plants are a rich source of phytochemical compounds, including glucosinolates and their hydrolysis products, phenolic compounds, carotenoids, and vitamin C [16]. Genetic composition is the main factor that directly influences the phytochemical composition of cabbage, as it determines their individual characteristics in terms of nutrient content, as well as traits such as color, shape, and size [17]. While various factors, including environmental conditions and developmental stage, impact the phytochemical profile and overall quality of Brassica species, genetic background is the primary factor affecting phytochemical content, and many breeding programs are aiming for the selection and creation of improved genotypes [18]. Several reviews on the Brassicaceae family have been conducted; however, there is limited information on the variation of phytochemical composition between different cultivars of cabbage, focusing on their morphological characteristics. The present review attempts to compile and compare available literature data of the last decade on the above research topic, with the ultimate aim of proposing and exploiting specific cultivars for breeding programs, as well as for the extraction and production of natural-based food pigments and dietary supplements.

## 2. Phytochemical Compounds

### 2.1. Glucosinolates 

Glucosinolates (GSLs) are sulfur-containing secondary metabolites found mainly in plants of the Brassicaceae family [19]. They are the major group of phytochemicals found in cabbage [20]. Their chemical structure consists of a β-D-glucopyranose unit and an O-sulfated anomeric (Z)-thiohydroximate function, linked to an aglycone with a variable side chain (R) (Figure 2) [21]. Based on the structure of their side chain and the type of precursor amino acid, GSLs are classified into aliphatic, indonyl, and aromatic GSLs [22]. When plant tissue containing glucosinolates is damaged, the endogenous enzymes β-thioglucosidases, also known as myrosinases, are released from cell vacuoles leading to the hydrolysis of these compounds [23]. The hydrolysis products of glucosinolates include isothiocyanates (ITCs), epithionitriles (EPTs), and nitriles, based on factors like pH, availability of ferrous ions, the compound’s side chain, and substrate type [24]. ITCs, the main products of GSL hydrolysis, are mostly responsible for the health-promoting properties of cabbage, such as reduced risk of cancer and cardiovascular disease [25].

When myrosinases are deactivated, e.g., when excess cooking occurs, intestinal bacteria exhibiting myrosinase activity that are present in the human distal gut are capable of hydrolyzing the ingested glucosinolates [26]. Since human gut bacteria is highly variable amongst individuals, so is glucosinolate metabolization into ITCs [27]. Following the production of ITCs, they are taken up by enterocytes through passive diffusion, and undergo spontaneous conjugation with glutathione. Conjugation is further promoted by glutathione-S-transferases (GSTs), including Mu (GSTM1), which is the major determinant of ITC metabolism, Pi (GSTP1), and Theta (GSTT1) [28]. The conjugate is then exported extracellularly by proteins MRP1 (multidrug resistance-associated protein-1), MRP2, and P-glycoprotein-1 (Pgp-1) [29,30]. Once exported to the extracellular matrix, ITC metabolites are transported though the hepatic portal vein, where they undergo metabolism via the mercapturic acid pathway. This pathway involves several enzymatic modifications, including cleavage of glutamine, resulting in cysteine–glycine conjugates, and cleavage of glycine, yielding cysteine conjugates, and finally acetylation, to form N-acetyl-cysteine (NAC) conjugates, which are then excreted in urine [31]. Free ITCs and their metabolites are also taken up by hepatocytes and subsequently enter the systemic circulation, allowing them to reach peripheral tissues like the breast and prostate, though their concentrations in these tissues are significantly lower than in the liver [25].

Over 20 ITCs have been identified as having anticarcinogenic effects against tumorigenesis [32]. These compounds inhibit cell proliferation by targeting proteins involved in tumor initiation and growth pathways [33]. Extensive epidemiological evidence suggests that consuming cruciferous vegetables is linked to a significantly reduced risk of developing cancer at various sites, including in the lungs [34], breast [35], bladder [36] and prostate [37]. In the case of prostate cancer in particular, diets rich in cruciferous vegetables have been linked to a reduced progression from localized to more aggressive forms [38].

Levels of GSLs vary greatly between different cultivars (Table 1). GSLs levels in white cabbage cultivars vary between 1.05 and 70.56 μmol/g d.w., while red cabbage cultivars show variation between 5.90 and 25.5 μmol/g d.w. [5,39]. Yue et al. (2024) compared the total glucosinolate content of different forms of cabbage (green spherical, green oblate, purple spherical, and green cow heart) and reported values of 1.32 μmol/g for green spherical form, 9.98 μmol/g for green oblate form, 15.05 μmol/g for green cow heart (cylindrical) form, and 10.54 μmol/g for purple spherical form, indicating that green cow heart varieties display higher glucosinolate content [40]. Oloyede et al. (2021) investigated the influence of cabbage accession, morphotype (including red, white, and Savoy cabbage), and growing condition on glucosinolates and their hydrolysis products, as well as myrosinase activity, concluding that the profile and concentration of GSLs and their hydrolysis products were significantly more affected by cabbage morphotype than by accession [41]. According to various researchers, the predominant GSLs in cabbage are aliphatic, followed by indonyl and aromatic [42,43,44].

In a study by Bhandari et al. (2020), the glucosinolate content of 146 different red and white cabbage cultivars was assessed to select suitable genotypes for a potential breeding program. The total glucosinolate content varied from 3.99 to 23.75 μmol/g d.w., and individual glucosinolates differed significantly among cultivars [44]. The selection of cultivars was based on the content of glucoabrassicin, glucoiberin, glucoraphanin, and sinigrin, as the hydrolysis of these GSLs display anticancer properties [23,45,46]. The highest glucoabrassicin content was observed in white cabbage cultivars BOL-AWS-1999-156, UR Gogetsu, Skvirskaya No 32, and Late Flat Dutch. White cabbage cultivars Podarok, Late Flat Dutch, Zuun kharaa No 15, and Zuun kharaa No 1 displayed the highest glucoiberin content (>4.0 μmol/g d.w.), while the highest sinigrin content was observed in white cabbage cultivars BOL-AWS-1999-153, Kashirka 202, Succession Green Leaved, Sagyahwak, Yujanka 31, and Zuun kharaa No 10 (>9.0 μmol/g d.w.). White cultivars Sudya, Sudiya-146, and TJK-PHJ-2014-6-8 had a higher glucoraphanin content (>3.0 μmol/g d.w.), but total GSL content was lower compared to other genotypes [44]. Researchers are interested in enhancing the glucoraphanin content in Brassica vegetables, as it is considered the most important GSL due to its health-promoting activities and anticancer properties [44,47,48]. Higher progoitrin content should also be taken into consideration, as it produces oxazolidine-2-thione, which is known to cause goiters and other harmful effects in mammals [44].

Robin et al. (2017) analyzed the glucosinolate profile of eight different cabbage genotypes, including commercial cultivars Rubra, YR gold, and Ohgane, and five inbred lines. A total of 11 GSLs were detected, ranging from 0.2 to 0.8 μmol/g d.w., of which seven were aliphatic (sinigrin, glucoiberin, glucoiberverin, glucoraphanic, gluconapin, glucoerucin, and progoitrin) and four were indolic (glucobrassicin, neoglucobrassicin, 4-methoxyglucobrassicin, and 4-hydroxyglucobrassicin). The GSL 4-methoxyglucobrassicin was found in all genotypes in trace amounts. Sinigrin was found in all genotypes apart from one inbred line, with the lowest amount (0.3 μmol/g f.w.) detected in green cabbage cultivar Ohgane. Glucoraphanin and gluconapin were detected only in cultivars Rubra and YR Gold. White cabbage cultivar YR Gold contained the highest total GSLs (157.6 μmol/g), which was significantly higher than the GSL content of red cabbage cultivar Rubra (46.8 μmol/g d.w.) and Ohgane (4.9 μmol/g d.w.) [49]. 

However, apart from genetic variation, several factors influence the levels of GLs in both white and red cabbage, including climate, region, harvest season, and developmental stage [50]. Schmidt et al. (2024) compared the total glucosinolate content of 26 cabbage cultivars with different agronomic characteristics (13 round white, 7 pointed white, 4 flat white, 2 Savoy, and 7 round red cultivars) grown in different seasons (summer, autumn and winter). Total content of GSLs was on average higher in winter cultivars compared to summer and autumn cultivars. The total GSL content varied significantly, from 38 to 242 mg/100 g f.w. (12 to 52 μmol/g d.w.). GSL content of Savoy (32.0–53.3 μmol/g d.w.) and red cabbage (21.9–44.4 μmol/g d.w.) cultivars were significantly higher than white cabbages (13.2–32.8 μmol/g d.w.). GSL composition also differed between different cabbage types. The major GSLs in white and Savoy cultivars were glucobrassicin, glucoiberin, and sinigrin, while red cabbage cultivars displayed high levels of progoitrin, gluconapin, and glucoraphanin [51]. Hanschen and Shreiner (2017) investigated the levels of GLs, as well as their hydrolysis products, in sprouts and fully developed heads of three white (Marcello, Perfecta, Tolsma), three red (Roodkop 2, Redma and Integro) and three Savoy (Capriccio, Daphne, Emerald) cabbage cultivars. According to their findings, sprouts almost always had higher GSL concentrations compared to fully developed heads. Among the white cabbage cultivars, the main GSL in both developmental stages was sinigrin, with cultivar Tolsma displaying the highest concentration (4.92 μmol/g f.w. in sprouts and 1.52 μmol/g f.w. in the fully developed head) compared to Perfecta (2.14 μmol/g f.w. in sprouts and 0.58 μmol/g f.w. in the fully developed head) and Marcello (2.36 μmol/g f.w. in sprouts and 0.68 μmol/g f.w. in the fully developed head). Cultivars Marcello and Perfecta were also rich in glucoiberin in both developmental stages, while the content of glucobrassicin was twice as high in fully developed heads of all three cultivars compared to sprouts. In red cabbage cultivars, the main GSL in both sprouts and fully developed heads was glucoraphanin, except for sprouts of Roodkop 2. Sprouts of Redma showed the highest concentrations of glucoraphanin (1.87 μmol/g f.w.), compared to Integro (1.04 μmol/g f.w.). Roodkop 2 sprouts had the highest concentration of sinigrin (1.29 μmol/g f.w.), compared to Redma (0.77 μmol/g f.w.) and Integro (0.95 μmol/g f.w.). Red cabbage sprouts were also rich in sinigrin, progoitrin, and glucoiberin, while fully developed heads contained considerable amounts of glucoabrassicin. The main GSLs of Savoy cabbage cultivars were sinigrin and glucoiberin. Higher concentrations of these GSLs were detected in cultivar Daphne, at concentrations of 5.35 μmol/g f.w. and 3.56 μmol/g f.w., respectively. Fully developed heads also contain glucobrassicin, which was the main GSL in cultivar Emerald (0.65 μmol/g f.w.) [52]. 

**Table 1 foods-13-03395-t001:** Glucosinolate content of different cabbage cultivars.

Cabbage Form	Cultivar	Total Glucosinolate Content		Reference
*Brassica oleraceae* var. *capitata* f. *alba*	Marcello	Sprouts	3.49 μmol/g f.w.	[51]
		Mature head	1.42 μmol/g f.w.	
	Perfecto	Sprouts	4.78 μmol/g f.w.	
		Mature head	1.85 μmol/g f.w.	
	Tolsma	Sprouts	5.18 μmol/g f.w.	
		Mature head	2.30 μmol/g f.w.	
	Ohgane	4.9 μmol/g d.w.		[49]
	YR Gold	157.6 μmol/g d.w.		
	BOL-AWS-1999-153	23.37 μmol/g d.w.		[44]
	BOL-AWS-1999-156	23.76 μmol/g d.w.		
	Late Flat Dutch	21.33 μmol/g d.w.		
	Podarok	19.49 μmol/g d.w.		
	Zuun Kharaa N 1	13.54 μmol/g d.w.		
	Zuun Kharaa N 10	19.39 μmol/g d.w.		
	Zuun Kharaa N 15	19.19 μmol/g d.w.		
	Kashirka 202	21.62 μmol/g d.w.		
	Succession Green Leaved	21.33 μmol/g d.w.		
	Sagyahwak	20.72 μmol/g d.w.		
	Yujanka 31	19.86 μmol/g d.w.		
	Valcatiecskaya	23.31 μmol/g d.w.		
	Natsuzoka	17.82 μmol/g d.w.		
	UR Gogetsu	17.66 μmol/g d.w.		
	Gyeongphong 1 ho	17.16 μmol/g d.w.		
	Skvirskaya N32	16.81 μmol/g d.w.		
	153	14.78 μmol/g d.w.		
	Tashkent 110	14.71 μmol/g d.w.		
	Golden Acre	11.32 μmol/g d.w.		
	Sudya	10.75 μmol/g d.w.		
	Sudiya-146	10.40 μmol/g d.w.		
	TJK-PHJ-2014-6-8	8.90 μmol/g d.w.		
*Brassica oleraceae* var. *capitata* f. *rubra*	Pourovo cervene	12.03 μmol/g d.w.		
	Kirmizi	18.18 μmol/g d.w.		
	Rubin	17.55 μmol/g d.w.		
	Red Drumhead 2	14.41 μmol/g d.w.		
	Integro	Sprouts	3.42 μmol/g f.w.	[52]
		Mature head	1.45 μmol/g f.w.	
	Redma	Sprouts	4.53 μmol/g f.w.	
		Mature head	1.02 μmol/g f.w.	
	Roodkop 2	Sprouts	2.92 μmol/g f.w.	
		Mature head	1.33 μmol/g f.w.	
	Rubra	46.8 μmol/g d.w.		[49]
*Brassica oleraceae* var. *capitata* f. *sabauda*	Capriccio	Sprouts	4.14 μmol/g f.w.	[51]
		Mature head	1.90 μmol/g f.w.	
	Daphne	Sprouts	9.61 μmol/g f.w.	
		Mature head	1.34 μmol/g f.w.	
	Emerald	Sprouts	3.46 μmol/g f.w.	
		Mature head	1.44 μmol/g f.w.	

### 2.2. Phenolic Compounds

Phenolic compounds are a diverse class of secondary metabolites, known to exhibit antioxidant, antimicrobial, and anti-inflammatory properties. They are compounds that contain a benzene ring that is substituted with a hydroxyl group known as phenol [53]. The main representatives of phenolic compounds in cabbage are flavonoids, mainly flavonols and anthocyanins, as well as hydroxycinnamic acids [54]. 

The antioxidant properties of phenolic compounds have garnered them considerable attention in recent years. Flavonoids possess strong antioxidant properties due to their ability to chelate with metals, inhibit the action of oxidases, and enhance the activity of antioxidant enzymes [55,56]. These compounds are also able to directly scavenge free radicals [55]. Hydroxycinnamic acids and their derivatives exhibit strong free radical scavenging properties and act as metal chelating agents [57,58]. Some hydroxycinnamic acids, such as caffeic acid, can also enhance the activity of endogenous antioxidant enzymes [59,60]. 

Phenolic compounds greatly vary between different forms of cabbage (Table 2). Total phenolic content in white cabbage cultivars varies between 24.83 and 60.36 mg GAE/g f.w., while red cabbage cultivars exhibit higher amounts, varying between 170.53 and 174.38 mg GAE/g f.w. [61]. Total phenolic content is higher in red cabbage cultivars, presumably due to the presence of anthocyanins [62,63]. Liang et al. (2019) compared the total phenolic content in different forms of cabbage commonly consumed in China and reported values of 125.54 mg GAE/100 g for white ball head cabbage, 119.34 mg GAE/100 g for white conical head cabbage, 86.64 mg GAE/100 g for white flat head cabbage, and 153.94 mg GAE/100 g for the purple spherical form [64]. Yue et al. (2024) investigated 142 green spherical, 8 green oblate, 6 purple spherical, and 3 green cow heart cabbage cultivars. The total phenolic content ranged from 3.1 to 18.48 mg/g in green spherical cultivars, 4.34–17.28 mg/g in green oblate cultivars, 3.48–15.85 mg/g in purple spherical cultivars and 8.85–13.3 mg/g in green cylindrical cultivars [40]. Singh et al. (2006) investigated the total phenolic content in ten different green cabbage cultivars (Gungaless, Pusa Mukta, Kirch-10, Resist Crown, Golden Acre, Quisto, Rare Ball, Mini Ball, Hari Rani Gol, Fieldman, Green Cornell, Green Yogendra, Green Challenger, and BC-76). Total phenolic content ranged from 12.6 mg GAE/100 g f.w. in Pusa Mukta to 34.4 mg GAE/100 g f.w. in Green Cornell [65]. Jakobek et al. (2018) and Loncaric et al. (2020) compared the total phenolic content of three traditional white cabbage varieties (Cepinski, Varazdinski, Ogulinski) and one commercial white cabbage hybrid (Bravo F1). The hybrid displayed lower phenolic content in both studies [66,67]. Bravo F1 was also studied by Voca et al. (2018) and displayed higher phenolic content (45.45 mg GAE/100 g f.w.) than white cabbage hybrids Bronco F1 (24.76 mg GAE/100 g f.w.) and Farao F1 (24.83 mg GAE/100 g f.w.) but lower than white cultivar Slava (60.36 mg GAE/100 g f.w.). The highest phenolic content was observed in red cabbage hybrids Maestro F1 and Primero F1 (174.38 and 170.53 mg GAE/100 g f.w., respectively) [61]. Leja et al. (2010) compared the total phenolic content of four red cabbage cultivars (Langedijker, Kissendrupm, Koda, Haco) and one white cabbage hybrid (Lennox F1), over two separate harvest years. The lowest phenolic content was observed in Lennox F1 (31.4–47.6 mg chlorogenic acid/100 g f.w.) and the highest in cultivars Kissendrup and Langedijker (213.2–288.3 mg chlorogenic acid/100 g f.w. and 248.5–273.2 mg chlorogenic acid/100 g f.w., respectively). A considerable variability among phenolic content was also observed in both years of the experiment [63]. Podsedek et al. (2006) evaluated the total phenolic content of two red (Kissendrup and Koda), three white (Almanag, Tukana, and Vestri) and two Savoy cabbage cultivars (Langedijker and 60F/100). The lowest phenolic content was observed in white cabbage cultivars, ranging from 20.81 mg GAE/100 g f.w. in cultivar Tukana to 29.70 mg GAE/100 g f.w. in cultivar Almanag. Highest phenolic content was observed in red cabbage cultivar Kissendrup (171.36 mg GAE/100 g) [68]. 

Hydroxycinnamic acids are the main constituent of non-flavonoid phenolic compounds in Brassica vegetables, including cabbage. The most common hydroxycinnamic acids found in Brassica vegetables are *p*-coumaric, ferulic, and sinapic acids (Figure 3), often found in conjugation with sugars or other hydroxycinammic acids [54]. Sinapic acid and its constituents are the main phenolic acids in various cultivars of both red and white cabbage [69]. According to Liang et al. (2019) [64], sinapic acid was the predominant phenolic acid in red (12,736.82 mg/g f.w.), flat head (787.15 mg/g f.w.), ball head (976.45 mg/g f.w.) and conical head cabbage (653.90 mg/g f.w.), followed by iso-ferulic acid ranging from 415.63 in conical head cabbage to 3482.20 mg/g f.w. in red head cabbage. Most phenolic acids in red head cabbage, with the exception of *p*-coumaric and chlorogenic acid, were significantly higher than in other the other forms of cabbage, particularly 3.5-dihydroxybenzoic acid, caffeic acid, ferulic acid, and sinapic acid, which were 10 to 40 times higher in red head cabbage compared to the other types [64]. As reported by Loncaric et al. (2020), sinapic acid was the main constituent of phenolic acids in white cabbage cultivars Cepinski (78.15 mg/kg), Varazdinski (96.45 mg/kg), Bravo F1 (68.56 mg/kg) and Ogulinski (65.9 mg/kg), followed by ferulic acid (6.52, 8.18, 4.96, and 7.77 mg/kg, respectively), and caffeic acid (1.76, 2.06, 1.49, and 2.16 mg/kg, respectively), while *p*-coumaric and chlorogenic acids were present in lower amounts [67]. According to a study by Fernandez-Leon (2014), sinapic acid was the main phenolic acid present in Savoy cabbage cultivar Dama (1.59 mg/100 g) but not in cultivar Leticia (0.46 mg/100 g), where the main phenolic acid present was chlorogenic acid (0.54 mg/100 g) [70]. Chlorogenic acid was also the main constituent of phenolic acids in Savoy cabbage, as reported by Martinez et al. (2010) [71]. 

Flavonols are the main flavonoid compounds found in both red and white cabbage, mainly *o*-glycosides of kaempeferol (10.2 and 15.4 μg/g d.w., respectively), quercetin (9.2 and 5.7 μg/g d.w., respectively), isorhamnetin (1.9 and 1.1 μg/g d.w., respectively) and myricetin (Figure 4), while the sugar part mainly consists of glucose, rhamnose, rutinose, galactose or arabinose [72]. Park et al. (2014) analyzed the phenolic content of nine white and red cabbage cultivars. Flavonols were only detected in red cabbage cultivars, mainly quercetin, followed by kaempeferol [69]. A study by Kim et al. (2004) revealed differences in flavonoid content with respect to quercetin and kaempeferol in 10 white cabbage cultivars (Bobcat, Fresco, Little rock, Marvelon, Rinda, Ramada, Transam, Genesee, Huron, and Octoking) at the juvenile stage. Quercetin was not detected in cultivars Genesee, Little Rock, or Huron, but ranged from 0.24 mg/100 g in cultivar Rinda to 1.46 in cultivar Bobcat. Kaempeferol was detected in all studied cultivars, ranging from 1.30 mg/100 g in Genesee to 7.03 mg/100 g in cultivar Rinda [73]. 

Red cabbage is one of the richest sources of anthocyanins in the plant kingdom (≥10 mg/g d.w.), however it highly depends on the cultivar, agricultural practices, and developmental stage [68,74]. Red cabbage extract contains more than 30 anthocyanins, mainly cyanidin-3-diglucoside-5 glucoside in acylated and non-acylated forms. The acyl groups are mainly *p*-coumaric, sinapic, and ferulic acids [75]. Ahmadiani et al. (2014) compared the anthocyanin content and profiles of seven red cabbage cultivars (Cairo, Kosaro, Integro, Buscaro, Azurro, Primero, and Bandolero) at two maturity stages (harvested 13 and 21 weeks after transplanting). According to their findings, cultivar had a significant impact on pigment profile, especially the proportions of monoacylated and diacylated pigment contents. Cultivars Primero and Azuro had the highest proportions of monoacylated pigments and lowest proportions of diacylated pigments, whereas Buscaro and Bandolero cultivars presented the lowest proportions of monoacylated and the highest proportions of diacylated pigments. Maturity also affected monoacylated and diacylated pigments differently, and the effect was highly dependent on the cultivar. Nonacylated anthocyanins showed a significant increase in the 21-week-mature plants, except for cultivars Azurro and Buscaro. Monoacylated anthocyanins decreased at 21 weeks, except for cultivars Primero and Azuro, whereas the proportion of diacylated pigments increased significantly in cultivars Kosaro and Buscaro at 21 weeks [76]. The number and type of acyl group affects the stability of anthocyanins. Studies have indicated that diacylated anthocyanins are more stable than monoacylated anthocyanins. The type of acyl group also affects their stability [77]. Acylation with sinapic acid reduces the stability of anthocyanins, compared to other hydroxycinnamic acids, but increases the antioxidant activity [78]. Diacylated anthocyanins also have higher antioxidant activity compared to monoacylated ones [79]. Therefore, it is important to choose the correct type of cultivars for application of red cabbage for dietary antioxidants or food colorants.

The phenolic profile and content, particularly flavonoids, are strongly linked to the antioxidant capacity of Brassica species, as phenolic compounds have shown greater antioxidant activity compared to vitamins and carotenoids [54]. Antioxidant capacity of flavonoids and phenolic acids depends on the number and arrangement of hydroxyl groups in their molecules. Presence of a catechol group in the B-ring, a 2,3-double bond in conjugation with a 4-oxo function in the C-ring, and hydroxyl groups at the 3 and 5 positions are the structural groups responsible for the free-radical scavenging and antioxidant activities of flavonoids. Quercetin possesses all three structural groups, making it a more potent antioxidant compared to kaempferol, which lacks the catechol moiety in the B-ring [80]. In phenolic acids, antioxidant activity is positively correlated with the number of hydroxyl groups attached to the aromatic ring, while the methoxy and carboxylic acid groups also affect influence the antioxidant capacity [81]. Antioxidant activity is therefore influenced by both the total content and the specific type of phenolic compounds, both of which are dependent on the specific cultivar.

### 2.3. Vitamin C

Vitamin C is an essential nutrient that cannot be synthetized by most vertebrates, including humans, due to lack of the enzyme that catalyzes the last step of its synthesis, L-gulonolactone oxidase [82]. The term Vitamin C refers to two separate compounds, ascorbic acid and dehydroascorbic acid. Dehydroscorbic acid is easily reduced in an organism to ascorbic acid, thus displaying the properties of vitamin C [83]. The most important biological function of ascorbic acid is its ability to stop the radical chain reaction [84]. In addition, it participates in the synthesis of collagen [85]. 

Singh et al. (2006) reported an average of 9.65 mg/100 g of vitamin C content in white, 24.38 mg/100 g f.w. in red, and 14.49 mg/100 g f.w. in Savoy cabbage [86]. According to Yue et al. (2024), vitamin C content ranged from 0.184 to 0.686 mg/g in green spherical cabbage cultivars, 0.164–0.528 mg/g in green oblate cultivars, 0.202–0.284 in purple cabbage, and 0.285–0.345 in green cow heart form [40]. Park et al. (2014) studied the vitamin C content of 38 inbred lines of white and 7 inbred lines of red cabbage. Vitamin C content in green cabbages ranged from 22.72 to 51.65 mg/100 g f.w., while red cabbages displayed variation between 36.57 and 129.90 mg/100 g f.w. [39]. 

While genetic background is the major factor determining vitamin C content, variations in the content of vitamin C can also be attributed to other factors [87]. Penas et al. (2011) investigated the vitamin C content of five white cabbage cultivars (Hinova, Megaton, Alfredo, Candela, and Bronco) grown in two different geographical regions of Spain. The vitamin C content of studied cultivars was dependent on the cultivar and the growing location. Among the cabbages cultivated in the north, cultivar Alfredo exhibited the highest vitamin C content (3.6 mg/g d.w.), followed by cultivars Hinova and Bronco (2.9 mg/g d.w.), Candela (2.7 mg/g d.w.), and Megaton (2.4 mg/g d.w.). Vitamin C content significantly increased in all cultivars when grown in the east, with values ranging from 4.2 to 6.0 mg/g d.w. Cultivar Candela displayed the highest vitamin C content (6.0 mg/g d.w.) followed by Hinova, Megaton, Alfredo, and Bronco (5.0, 4.3, 5.5, 5.6 mg/g d.w., respectively) [88]. 

### 2.4. Carotenoids

Carotenoids are tetraterpene pigments that exhibit yellow, orange, red, and purple colors and are widely distributed in photosynthetic bacteria, some species of archaea and fungi, algae, and plants [89]. Their general structure consists of a polyene chain with nine conjugated double bonds and an end group at both ends of their chain. The conjugated double bonds are primarily responsible for their pigmenting properties and their ability to interact with free radicals and act as effective antioxidants [90]. Carotenoids are divided into two groups, the non-oxygenated carotenes and oxygenated xanthophylls. Carotenes, such as a-carotene, β-carotene, and lycopene, are hydrocarbon, while xanthophylls such as β-cryptoxanthin, lutein, and zeaxanthin contain oxygen atoms as hydroxy, carbonyl, aldehyde, carboxylic, epoxide, and furanoxide groups in their molecules [89]. The primary carotenoids in *Brassica* species are lutein and β-carotene, but at least 16 carotenoids have been identified [68]. According to Voca et al. (2018), carotenoid content varied from 0.15 mg/g in white cabbage cultivar Bronco F1 to 0.42 mg/g in white cabbage cultivar Slava, while red cabbage cultivars Farao F1 and Maestro F1 did not contain any carotenoids [61]. Podsedek et al. (2006) reported values of 0.014–0.016 mg/100 g f.w. in red cabbage cultivars, 0.009–0.042 mg/100 g f.w. in white cabbage cultivars, and 0.048–0.122 mg/100 g f.w. in Savoy cabbage cultivars [68]. Singh et al. (2006) investigated the contents of β-carotene and Lutein in white, red, and Savoy cabbage, reporting values of 0.050, 0.044, and 0.074 mg/100 g f.w., respectively [86]. 

## 3. Conclusions

Cabbage is an important source of bioactive compounds, known since ancient times for its health promoting properties. Based on this review, it can be concluded that the phytochemical composition of cabbage is significantly affected by its form and cultivar. Many studies have shown that there are significant differences between different cabbage forms, each of them displaying certain advantages in terms of phytochemical types and subsequent applications. Red cabbage generally displays higher phenolic content, particularly due to the presence of anthocyanins, though their profile and stability is highly dependent on the specific cultivar. Choosing the correct type of cultivar is therefore important for the production of high quality, stable natural food colorants. On the other hand, white cabbages tend to have a higher GSL content, mainly influenced by the cultivar. Therefore, it was particularly important to conduct a review considering the form and cultivar of cabbage for future applications, including breeding programs, production of natural food pigments, and dietary supplements. Further research in this domain would integrate the application of smart robotics (artificial intelligence, automation, biotechnology and advanced analytics) for the extraction of phytochemicals from plants in order to improve the efficiency, the precision, the sustainability, and scalability of the process.

## Figures and Tables

**Figure 1 foods-13-03395-f001:**
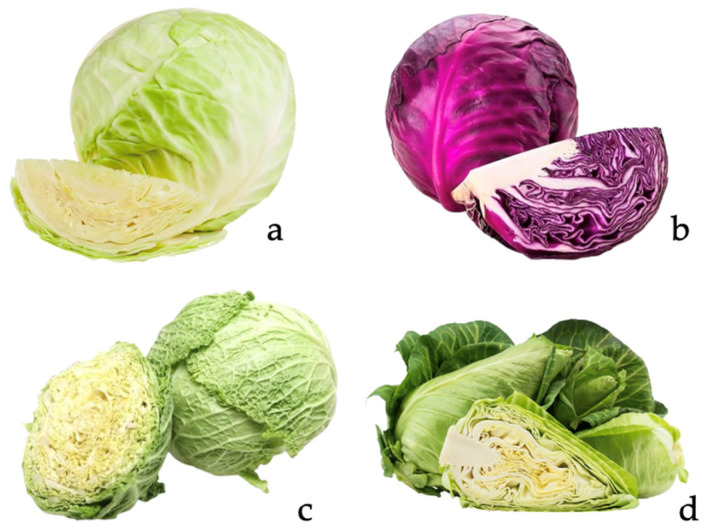
Different forms of cabbage. (**a**) *Brassica oleraceae* var. *capitata *f. *alba*, (**b**) *Brassica oleaceae* var. *capitata* f. *rubra*, (**c**) *Brassica oleraceae* var. *capitata* f. *sabauda*, (**d**) *Brassica oleraceae* var. *capitata* f. *acuta*.

**Figure 2 foods-13-03395-f002:**
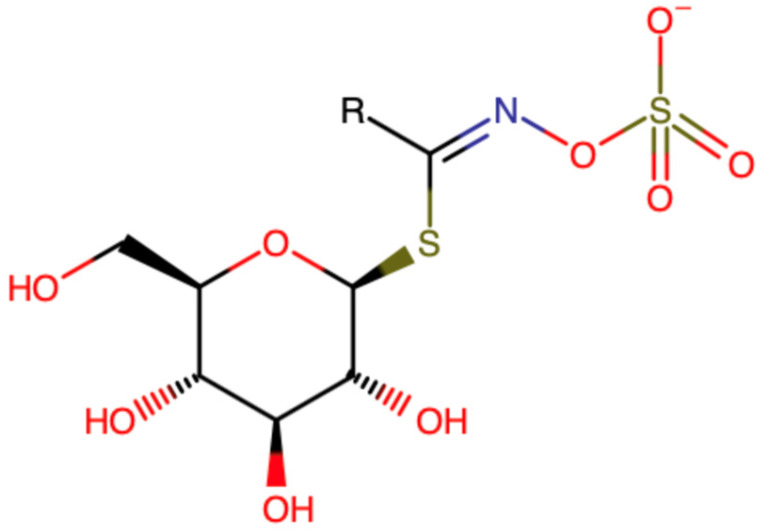
Generic structure of glucosinolate. The side group R varies.

**Figure 3 foods-13-03395-f003:**
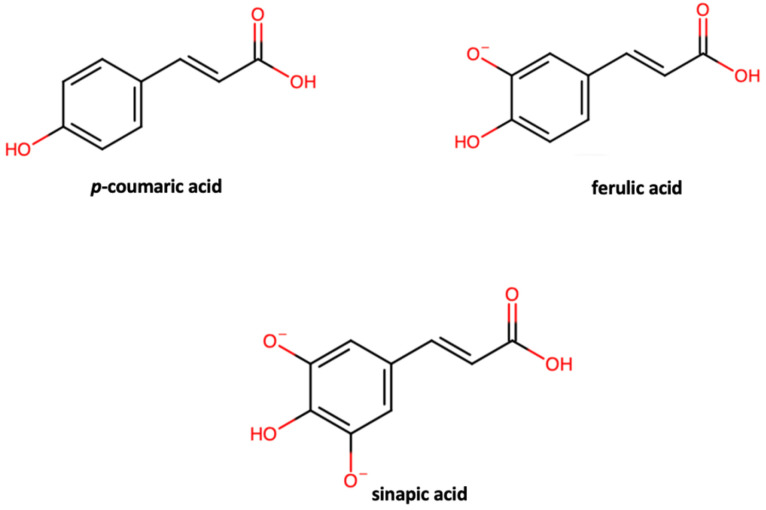
Chemical structures of most common hydroxycinnamic acids in cabbage.

**Figure 4 foods-13-03395-f004:**
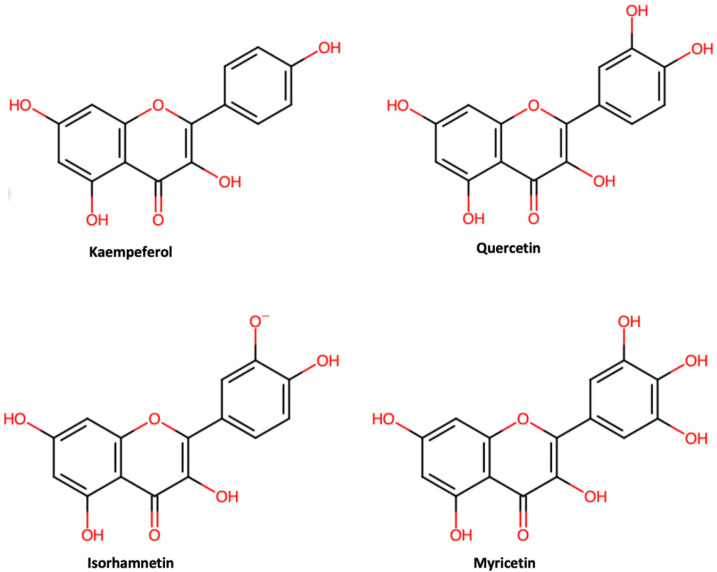
Chemical structures of most common flavonols in cabbage.

**Table 2 foods-13-03395-t002:** Total phenolic content of different cabbage cultivars.

Cabbage Form	Cultivar	Total Phenolic Content	Reference
*Brassica oleraceae* var. *capitata* f. *alba*	Bravo F1	45.45 ± 0.26 mg GAE/100 g f.w.	[61]
		466.7 ± 35.6 mg GAE/kg f.w.	[66]
		401.62 ± 11.10 mg GAE/kg f.w.	[67]
	Bronco F1	24.76 ± 0.46 mg GAE/100 g f.w.	[61]
	Slava	60.46 ± 0.41 mg GAE/100 g f.w.	
	Farao F1	24.83 ± 0.7 mg GAE/100 g f.w.	
	Cepinski	564.9 ± 67.1 mg GAE/kg f.w.	[66]
		496.0 ± 25.88 mg GAE/kg f.w.	[67]
	Varazdinski	598.2 ± 74.7 mg GAE/kg f.w.	[66]
		494.80 ± 7.43 mg GAE/kg f.w.	[67]
	Ogulinski	538.3 ± 46.7 mg GAE/kg f.w.	[66]
		480.95 ± 1.40 mg GAE/kg f.w.	[67]
	Gungaless	15.4 mg GAE/100 g f.w.	[65]
	Pusa Mukta	12.6 mg GAE/100 g f.w.	
	Kirch-10	18.1 mg GAE/100 g f.w.	
	Resist Crown	27.1 mg GAE/100 g f.w.	
	Golden Acre	13.1 mg GAE/100 g f.w.	
	Quisto	31.0 mg GAE/100 g f.w.	
	Rare Ball	18.2 mg GAE/100 g f.w.	
	Mini Ball	16.4 mg GAE/100 g f.w.	
	Hani Rari Gol	15.1 mg GAE/100 g f.w.	
	Fieldman	18.7 mg GAE/100 g f.w.	
	Green Cornell	34.4 mg GAE/100 g f.w.	
	Green Yogendra	15.7 mg GAE/100 g f.w.	
	Green Challenger	13.7 mg GAE/100 g f.w.	
	BC-76	12.9 mg GAE/100 g f.w.	
	Lennox F1	31.4–47.6 mg chlorogenic acid/100 g f.w.	[63]
*Brassica oleraceae* var. *capitata* f. *rubra*	Langedijker	248.8–273.2 mg chlorogenic acid/100 g f.w.	
	Tukana	20.81 ± 0.79 mg GAE/100 g f.w.	[68]
	Vestri	23.32 ± 0.47 mg GAE/100 g f.w.	
	Almanag	29.70 ± 0.66 mg GAE/100 g f.w.	
	Kissendrup	171.36 ± 13.77 mg GAE/100 g f.w.	
		201.5–288.3 mg chlorogenic acid/100 g f.w.	[63]
	Koda	134.73 mg GAE/100 g f.w.	[68]
		114.1–209.6 mg chlorogenic acid/100 g f.w.	[63]
	Haco	177.1–215.5 mg chlorogenic acid/100 g f.w.	
*Brassica oleraceae* var. *capitata* f. *sabauda*	Langedijker	54.31 mg GAE/100 g f.w.	[68]
	60F/100	47.62 mg GAE/100 g f.w.	

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
