# Peer review of "Overview of Phytochemical Composition of *Brassica oleraceae* var. *capitata* Cultivars"

_foods, 2024, doi:10.3390/foods13213395_

Round 1

Reviewer 1 Report

Comments and Suggestions for Authors

The manuscript (foods-3262002-peer-review-v1) focused on the variation of phytochemical composition and antioxidant activity of the Brassicaceae family. The phytochemical composition of many species has been widely investigated and shown health promoting. Overall, the idea of review about the phytochemical composition seems to be interesting. The manuscript is well written and focused. However, some issues are suggested to be addressed.

1.      What is the name of the gene about phenolic compounds?

2.      In the manuscript, the authors focused on the genetic background. How do the genes affect the phenolic content?

3.      Which factors affect the expression of the gene? Which parts have a high content of phenolic compounds? Which processing methods are easy to digest and absorb of phenolic compounds?

Please explain the above questions to facilitate a deeper understanding of the genetic background for readers.

Author Response

Point to Point answers to Reviewer's 1 comments      

Comments and Suggestions for Authors

The manuscript (foods-3262002-peer-review-v1) focused on the variation of phytochemical composition and antioxidant activity of the Brassicaceae family. The phytochemical composition of many species has been widely investigated and shown health promoting. Overall, the idea of review about the phytochemical composition seems to be interesting. The manuscript is well written and focused. However, some issues are suggested to be addressed.

  1. What is the name of the gene about phenolic compounds?

Response: Some of the genes involved in the phenolic acid and flavonoid biosynthesis pathways include the phenylalanine ammonia-lyase (PAL) gene family, chalcone synthase 3 (CHS3), cinnamyl alcohol dehydrogenase 9 (CAD9), CYP73A5 and 4-coumarate CoA ligase 4 (4CL) [ Yang, X.; Liao, X.; Yu, L.; Rao, S.; Chen, Q.; Zhu, Z.; Cong, X.; Zhang, W.; Ye, J.; Cheng, S.; et al. Combined Metabolome and Transcriptome Analysis Reveal the Mechanism of Selenate Influence on the Growth and Quality of Cabbage (Brassica Oleracea Var. Capitata L.). Food Res. Int. 2022, 156, 111135, doi:10.1016/j.foodres.2022.111135]. Those genes encode different enzymes involved in the shikimate and phenylpropanoid pathways, which are responsible for the production of phenolic compounds, including phenolic acids and flavonoids.

  1. In the manuscript, the authors focused on the genetic background. How do the genes affect the phenolic content?

Response: Genetic background is defined as ‘the entire genetic and genomic context of an organism; the complete genotype of an organism across all loci’. While, to our knowledge, transcriptomic analysis of cabbage cultivars, regarding the expression of genes involved in the production of phenolic compounds has not yet been conducted, gene expression typically plays an essential role in polyphenol production. For example, when it comes to cabbage cultivars of different color (red and white), genes involved in the flavonoid pathways are expressed differently in part due to the presence of anthocyanins, a type of flavonoid, in red cultivars. It would certainly be useful to carry out a transcriptomic analysis of the different phenotypes of cabbage with regard to flavonoid pathways; however, it was out of the scope of our review.

  1. Which factors affect the expression of the gene? Which parts have a high content of phenolic compounds? Which processing methods are easy to digest and absorb of phenolic compounds?

Response: Factors that affect the expression of the gene are typically related to the color of the cultivar, which is inherently a genetic factor, as well as environmental conditions and developmental stage (as mentioned on Lines 56-59 of the revised manuscript). Different plant tissues also have varying levels of phenolic compounds. Outer leaf layers typically contain more phenolic compounds in cabbage compared to inner layers. Phenolic compounds in plants can be linked to cell membranes and walls, therefore food processing methods such as utilization of high temperatures or freezing could potentially increase their bioavailability in the human body.

Please explain the above questions to facilitate a deeper understanding of the genetic background for readers.

Response: We thank the Reviewer for his/her input. Phenolic compound production at a genetic level is a very interesting, but complicated matter. A deeper analysis of genes and pathways involved in the production of phenolic compounds is out of the scope of this review. We have, however, taken the Reviewer’s input into consideration, as interesting points, concerning the gene expression in cabbage, were raised that require further research in the future.

Reviewer 2 Report

Comments and Suggestions for Authors

The article Overview of phythochemical composition of Brassica oleraceae var. capitata cultivars presents the differences in phytochemical composition between different cultivars of cabbage, focusing on their morphological differences.

The topic of the manuscript is relevant to the Journal and summarized data provided are important for breeding programs, production of natural food pigments and dietary supplements.

It should be noted that the abstract of the manuscript do not provide any quantitative data and not clear aim of the study is formulated.

The introduction section is well stricture, however reference 18 [L-52] rather deals with environmental conditions that influence the level of photochemical in Brassica vegetables, emphasizing the effect of different agronomic practices, although they are strictly genotype-dependent.

In the review article no methodology is mentioned and it is not clear how the literature data have been selected to be referred in the overview, which results in further discrepancies in the references, provided in Tables and in the narrative.

Section 2.1 provides thorough overview of glucosinolates structure and metabolism, along with data for their content in different cabbage cultivars. However, references for Total glucosinolate content in Table 1 are more related with review articles about Crusiferous vegetable intake and cancer risk, eg. References [ 34, 36 ].  In the body text of this section many other literature sources are also mentioned and it is not evident why some of the referred data are in tabular form and other are not.

Section 2.2 deals with Phenolic compounds. Their antioxidant properties and health benefits are well described. Regarding, provided data about Total phenolic content, it should be noted that this general term is related with Total Phenolics determined by spectrophotometric analysis with Folin-Ciocalteo reagent method and the results are generally expressed as mg GAE/g, 100g, or kg. in this respect, literature source 43, included in Table 2, provides data for Total anthocyanin content in 25 lines of red cabbage expressed in mg/g d.f. In the manuscript in Table 2, reference 43 – data for Total phenolic content expressed as mg/100 g f.w. are reported, which do not correspond with the literate source, neither with the commonly accepted definition of Total Phenolic content in the analytical practice.

Other point is that Cultivars selected in Table 1 and 2 are different and it is difficult to follow the level of photochemical composition presented in Brassica oleraceae var. capitata cultivars.

The article provides no clear conclusions that could emphasize the novelty of the work.

Taking into consideration the abovementioned discrepancies, the manuscript, it is suitable after major corrections.

Author Response

Point to Point answers to Reviewer's 2 comments    

Comments and Suggestions for Authors

  1. The article Overview of phythochemical composition of Brassica oleraceae var. capitata cultivars presents the differences in phytochemical composition between different cultivars of cabbage, focusing on their morphological differences.

The topic of the manuscript is relevant to the Journal and summarized data provided are important for breeding programs, production of natural food pigments and dietary supplements.

Response: We thank the Reviewer for his/her nice comments.

  1. It should be noted that the abstract of the manuscript do not provide any quantitative data and not clear aim of the study is formulated.

Response: The abstract was revised accordingly.

  1. The introduction section is well stricture, however reference 18 [L-52] rather deals with environmental conditions that influence the level of photochemical in Brassica vegetables, emphasizing the effect of different agronomic practices, although they are strictly genotype-dependent.

Response: We thank the Reviewer for his/her suggestion. Indeed, according to reference 18, mainly genotype characteristics, and subsequently environmental and agricultural factors, influence the phytochemical composition of Brassica plants. The sentence was revised accordingly.

  1. In the review article no methodology is mentioned and it is not clear how the literature data have been selected to be referred in the overview, which results in further discrepancies in the references, provided in Tables and in the narrative.

Response: Literature data was selected based on the last decade research articles of the topic. The discrepancies have been dealt with and Introduction section was revised accordingly.

  1. Section 2.1 provides thorough overview of glucosinolates structure and metabolism, along with data for their content in different cabbage cultivars. However, references for Total glucosinolate content in Table 1 are more related with review articles about Crusiferous vegetable intake and cancer risk, eg. References [ 34, 36 ]. In the body text of this section many other literature sources are also mentioned and it is not evident why some of the referred data are in tabular form and other are not.

Response: We thank the Reviewer for noticing the discrepancies. The references in Table 1 have been revised and updated.

  1. Section 2.2 deals with Phenolic compounds. Their antioxidant properties and health benefits are well described. Regarding, provided data about Total phenolic content, it should be noted that this general term is related with Total Phenolics determined by spectrophotometric analysis with Folin-Ciocalteo reagent method and the results are generally expressed as mg GAE/g, 100g, or kg. in this respect, literature source 43, included in Table 2, provides data for Total anthocyanin content in 25 lines of red cabbage expressed in mg/g d.f. In the manuscript in Table 2, reference 43 – data for Total phenolic content expressed as mg/100 g f.w. are reported, which do not correspond with the literate source, neither with the commonly accepted definition of Total Phenolic content in the analytical practice.

Response: We thank the Reviewer for noticing the discrepancies. The references in Table 2 and the results have been revised and updated.

  1. Other point is that Cultivars selected in Table 1 and 2 are different and it is difficult to follow the level of photochemical composition presented in Brassica oleraceae var. capitata cultivars.

Response: Table 1 focuses on total glucosinolate (GSL) content, whereas Table 2 presents total phenolic content of cultivars that have been reported in literature. Moreover, a discussion on GSL content of various cultivars was added in Lines 184-194.  

  1. The article provides no clear conclusions that could emphasize the novelty of the work. Taking into consideration the abovementioned discrepancies, the manuscript, it is suitable after major corrections.

Response: Conclusion section was revised according to Reviewer suggestions.

Reviewer 3 Report

Comments and Suggestions for Authors

The manuscript titled "Overview of phytochemical composition of Brassica oleraceae var. capitata cultivars" provides a comprehensive review of the phytochemical composition, particularly glucosinolates, phenolic compounds, anthocyanins, vitamin C, and carotenoids, in different cultivars of cabbage (Brassica oleraceae var. capitata). Overall, the manuscript provides a valuable overview of the phytochemical composition of different cabbage cultivars and its implications for health and agriculture. With the following suggestions, the manuscript could be further strengthened and made more impactful in the field of food science and nutrition.

1. The manuscript would benefit from a more detailed comparison of the phytochemical profiles between different cabbage cultivars, especially in the context of their antioxidant activities. While the authors have mentioned the variations, a deeper analysis or discussion on how these variations affect the overall antioxidant capacity of each cultivar would be valuable.

2. The review could be strengthened by including a section on the impact of postharvest storage and processing on the phytochemical content of cabbage. This information is crucial for understanding the stability of these compounds and their potential bioavailability in human diets.

3. It is important to note that while the manuscript focuses on the health-promoting properties of cabbage, a balanced discussion should also consider any potential antinutritional factors or adverse effects associated with high consumption of certain glucosinolate breakdown products.

Author Response

Point to Point answers to Reviewer's 3 comments    

Comments and Suggestions for Authors

The manuscript titled "Overview of phytochemical composition of Brassica oleraceae var. capitata cultivars" provides a comprehensive review of the phytochemical composition, particularly glucosinolates, phenolic compounds, anthocyanins, vitamin C, and carotenoids, in different cultivars of cabbage (Brassica oleraceae var. capitata). Overall, the manuscript provides a valuable overview of the phytochemical composition of different cabbage cultivars and its implications for health and agriculture. With the following suggestions, the manuscript could be further strengthened and made more impactful in the field of food science and nutrition.

  1. The manuscript would benefit from a more detailed comparison of the phytochemical profiles between different cabbage cultivars, especially in the context of their antioxidant activities. While the authors have mentioned the variations, a deeper analysis or discussion on how these variations affect the overall antioxidant capacity of each cultivar would be valuable.

Response: We thank the Reviewer for his/her suggestions. We have modified the manuscript accordingly (Lines 372-385 of the revised manuscript).

  1. The review could be strengthened by including a section on the impact of postharvest storage and processing on the phytochemical content of cabbage. This information is crucial for understanding the stability of these compounds and their potential bioavailability in human diets.

Response: Post-harvest storage and processing affect the phytochemical content of cabbage in different ways. Optimal storage conditions, drying, fermentation and other processes could be utilized to preserve phytochemical content. Authors consider, however, that an in-depth analysis of how the abovementioned factors affect phytochemical composition could be a subject of a future review.

  1. It is important to note that while the manuscript focuses on the health-promoting properties of cabbage, a balanced discussion should also consider any potential antinutritional factors or adverse effects associated with high consumption of certain glucosinolate breakdown products.

Response: We thank the Reviewer for his/her meaningful suggestion. We have revised the manuscript according to Reviewer’s suggestion (Lines 167-169 of the revised manuscript). 

Reviewer 4 Report

Comments and Suggestions for Authors

Explain in detail the novelty of this work. Include the differences between this research and any related work in the past. What research gaps were addressed in this research?

It is important to state clearly the implications for research, practice, and society. In my opinion, for the audience, for those who are not working in this field, but are interested in this subject (nutritionists, education specialists, public health organizations, different organizations, policymakers, and food industries), it is relevant to emphasize the importance of bioactive compounds from vegetables in the diet.  The authors are highly advised to add a short section (to the introduction ), for this purpose, the following references may be checked: DOI: http://dx.doi.org/10.5772/intechopen.91218

A few tables and figures should be included related to the topic.

Automation technologies are critical to reducing the time and cost of bioactive compound extraction and food product innovation. The absence of discussions about using sensors, data analytics, or smart robotics in the food industry is a gap that could modernize the field of food processing. You may write a few lines to highlight its significance.
Avoid old references and include at least 60% citations from 2023 and 2024.

The authors should include a Graphic abstract that describes the work carried out and the research outcome. During research what are all the methodologies adopted and developed things need to be incorporated in the graphical abstract?

Conclusion - In this section, authors should summarize the results and limitations of their study, as well as the scope for future research.

Author Response

Point to Point answers to Reviewer's 4 comments   

Comments and Suggestions for Authors

  1. Explain in detail the novelty of this work. Include the differences between this research and any related work in the past. What research gaps were addressed in this research?

Response: The present review analyzes the most important phytochemical compounds present in cabbage, focusing on the variation of phytochemical composition and antioxidant activity among different cultivars and emphasizing on their morphological characteristics.

  1. It is important to state clearly the implications for research, practice, and society. In my opinion, for the audience, for those who are not working in this field, but are interested in this subject (nutritionists, education specialists, public health organizations, different organizations, policymakers, and food industries), it is relevant to emphasize the importance of bioactive compounds from vegetables in the diet. The authors are highly advised to add a short section (to the introduction ), for this purpose, the following references may be checked: DOI: http://dx.doi.org/10.5772/intechopen.91218. A few tables and figures should be included related to the topic.

Response: We thank the Reviewer for his/her valuable suggestion. Introduction was revised according to Reviewer’s comments.

  1. Automation technologies are critical to reducing the time and cost of bioactive compound extraction and food product innovation. The absence of discussions about using sensors, data analytics, or smart robotics in the food industry is a gap that could modernize the field of food processing. You may write a few lines to highlight its significance. Avoid old references and include at least 60% citations from 2023 and 2024.

Response: We thank the Reviewer for his/her valuable and interesting suggestion. Authors consider that it would be an important topic for further research. Conclusion section was revised accordingly. As regards the references, authors have included research articles, reviews and book chapters from a wider range of years due to the specification of topic and the amplitude of information.

  1. The authors should include a Graphic abstract that describes the work carried out and the research outcome. During research what are all the methodologies adopted and developed things need to be incorporated in the graphical abstract?

Response: We thank the Reviewer for his comment; however graphic abstract is not mandatory. The methodology adopted has been included in the Introduction section.

  1. Conclusion - In this section, authors should summarize the results and limitations of their study, as well as the scope for future research.

Response: The results of the study are summarized in the Conclusion section and demonstrate that it is important to consider the form and cultivar of cabbage for future applications, including breeding programs, production of natural food pigments and dietary supplements. Moreover, the scope for future research was added.

Round 2

Reviewer 4 Report

Comments and Suggestions for Authors

The authors have made partial improvements to the paper based on previous feedback. I have no further comments.